Effects of coconut water on blood sugar and retina of rats with diabetes

Dai Yanan 1
Peng Li 2
Zhang Xiaohua 1
Wu Qingjing 1
Yao Jie 1
Xing Qiu 1
Zheng Yunyan 1
Huang Xiaobo 1
Chen Shaomei 1
Xie Qing opht_xq@163.com 1
1 Department of Ophthalmology, Central South University Xiangya School of Medicine Affiliated Haikou Hospital, Haikou, Hainan, P.R. China , Haikou , China
2 Department of Ophthalmology, The Second Xiangya Hospital, Central South University, Changsha, Hunan , Changsha , China
Ing Edsel
Electronic publication date: 2021 Jan 29
Publication date: 2021
Volume: 9
Electronic Location ID: e10667
Received 2020 Apr 14; Accepted 2020 Dec 8
Copyright: ©2021 Dai et al.
Copyright year: 2021
Copyright holder: Dai et al.
License: This is an open access article distributed under the terms of the Creative Commons Attribution License, which permits unrestricted use, distribution, reproduction and adaptation in any medium and for any purpose provided that it is properly attributed. For attribution, the original author(s), title, publication source (PeerJ) and either DOI or URL of the article must be cited.
License URL: https://creativecommons.org/licenses/by/4.0/

Keywords: Coconut water, Hypoglycemia, Diabetic retinopathy, Diabetic rats, Vascular endothelial growth factor

Funding: The Key R & D Program of Hainan Province in 2017 ZDYF2018235 This study is supported by the Key R & D Program of Hainan Province in 2017 (grant number ZDYF2018235). The funders had no role in study design, data collection and analysis, decision to publish, or preparation of the manuscript.

==============================
Background

This study aimed to evaluate the effects of coconut water on the general condition (fasting blood sugar and body weight) and retina of diabetic rats.

Methods

Forty-eight Sprague–Dawley male rats were divided into normal controls (NC), diabetes mellitus (DM), diabetes+coconut water (DM+CW), and diabetes+glibenclamide (DM+Gli) groups. After 4 weeks of normal feeding, coconut water was given to group III-DM+CW and 0.6 mg/kg glibenclamide to group IV–DM+Gli. The blood sugar, body weight, total retinal thickness, pathological changes, and VEGF expression in the retina were analyzed at different time points.

Results

The fasting blood sugar was 4–6 mmol/L in group I-NC and continuously increased in group II-DM, whereas gradually decreased after the 4th experiment week in the remaining two groups. The rats, except in group I-NC, have lost weight. In group II-DM, the total retinal thickness was significantly increased after the 8th and 12th experiment week, and the pathological changes in retina were observed. VEGF was almost fully expressed in the ganglion cell layer and inner granular layer and partially expressed in the outer granular layer in group II-DM, and mainly expressed in the ganglion cell layer and inner layer in group I-NC, with a lighter color. Group III-DM + CW and group IV-DM + Gli demonstrated similar VEGF expression as in group I-NC.

Conclusions

Coconut water has the potential to reduce blood sugar and diabetic retinal damage, serving as a candidate drug or nutrient for treating diabetes and its complications.

Introduction

Diabetes is one of the most significant public health problems worldwide. According to the International Diabetes Fedaration (2014), there were 382 million diabetics worldwide in 2013 with a projected 592 million in 2035. Diabetes mellitus (DM) causes acute metabolic disorder syndrome, leading to long-term damage, dysfunction, and failure of some organs due to chronic hyperglycemia, and may lead to death. Among these complications, diabetic retinopathy (DR) is one of the predominant microvascular complications of type 1 or type 2 DM and is a major ocular complication. Developed countries, including Europe and America, have conducted a number of epidemiological studies on diabetic retinopathy. According to the Eye Diseases Prevalence Research Group, an American research organization, the incidence of diabetic retinopathy can be as high as 40.3% (Kempen et al., 2004). According to the National Health and Nutrition Examination Survey, the incidence rate of diabetic retinopathy in patients over 40 years old was 28.5% (Zhang et al., 2010). The incidence of DR in Chinese patients with DM is approximately 23.0%. The visual impact of proliferative retinopathy has serious, potentially negative implications for the employment and life of diabetic patients (Xu et al., 2020).

Hyperglycemia is a classic biochemical abnormality in diabetes. It can cause four metabolic pathway disorders, which include the activation of the polyol pathway, the activation of the hexosamine biosynthesis pathway (HBP), the over-modification of protein by N-acetylglucosamine, and the activation of protein kinase C (PKC) leading to the formation of the advanced glycation end product. Disorderly regulation of these biochemical mechanisms at the molecular level is the main pathogenesis of DR. The current clinical treatments tend to focus on diabetic macular edema and proliferative diabetic retinopathy. These conditions are treated with intravitreal applications of various VEGF antagonists, corticosteroids, and vitreous surgery. The results of these treatments are unsatisfactory and expensive and they do not prevent or reverse the progression of diabetic retinopathy.

Early intervention is important to prevent or delay the development of DR, and to prevent and treat blindness secondary to this disease. Effective drug treatments for early diabetic retinopathy are still being explored.

Coconut water (CW) is a natural beverage obtained from the cavity of the coconut fruit. Recent animal experiments have reported that CW can reduce blood glucose levels, regulate carbohydrate metabolism, and improve antioxidant capacity. Several studies have reported that CW may alleviate kidney damage caused by diabetes (Nwangwa, 2012; Pinto et al., 2015; Nwangwa, 2012; Prathapan & Rajamohan, 2011; Preetha et al., 2012; Preetha, Girija Devi & Rajamohan, 2013). Both DR and diabetic nephropathy are retinal microangiopathic diseases and it is unknown whether CW can alleviate diabetic retinal damage.

Glibenclamide is a hypoglycemic medication that stimulates pancreatic beta cells. Its availability and affordability makes glibenclamide popular for the management of type 2 diabetes mellitus (Rambiritch et al., 2007). We sought to evaluate the effects of CW on the fasting blood glucose, body weight, and retinal protection of diabetic rats. We also explored the usefulness of natural drugs in the treatment of DR.

Materials & Methods

Animals

We studied 48 10-week-old male Sprague-Dawley (SD) rats, weighing 350 ± 30 g with clear ocular media and no ocular fundal lesions when examined by slit lamp microscope and ophthalmoscope. The rats were kept in a pathogen-free facility with access to food and water under controlled conditions (temperature 18−26 °C; humidity 40–70%; and 12-h light/dark cycle).

All SD rats and standard feed were purchased from Changsha Tianqin Biotechnology Co., Ltd. (SCXK, Xiang, 2014-0011). All SD rats and their feeds were tested physically and chemically by the Center for Disease Control and Prevention of Hunan Province (sample acceptance Nos. 2017DW041, No.2017DS009). All experimental procedures were approved by the biomedical ethics committee of Haikou People’s Hospital. (2017-087), and all efforts were made to minimize the suffering of the test subjects.

Local mature coconuts (6–8 months old) were purchased weekly in Wenchang, Hainan to ensure the consistency of the CW. The CW was collected and stored in a refrigerator at 4 °C for experimentation. We obtained the following materials for use in our study: Glibenclamide (H12020790; Tianjin Pacific Pharmaceutical Co., Ltd.); Streptozotocin (S0130; Beijing Sein Tan Technology Co., Ltd); citric acid–sodium citrate buffer (pH 4.5) (Xiamen Sea Standard Technology Co., Ltd); ketamine (Daiichi-Sankyo, Co., Ltd., Tokyo, Japan); xylazine (Bayer AG, Leverkusen, Germany); rabbit anti-mouse VEGF antibody (p80084, PhD Bio Company); fast enzyme-labeled sheep anti-mouse/rabbit IgG polymer (kit-5930, Fuzhou Maixin Biotechnology Development Co., Ltd.); slit lamp microscope (SL-3G, Topcon); ophthalmology (YZ11D, Suzhou Liuliu Vision Technology Co., Ltd.); FreeStyle Optium H (Abbott Laboratories, Chicago, IL, USA); Optical coherence tomography [(SPECT) RALIS OCT, Heidelberg, Germany]; electronic balance (XPE105; METTLER TOLED O, Switzerland); and a full automatic HE dyeing machine (Coverstainer, Dako, Denmark).

Diabetes model

All male SD rats were fasted for 12 h but had free access to water. Twelve rats were randomly selected for group I-NC using the random number table method. Sodium citrate buffer (pH 4.5) was injected intraperitoneally at a dose of 60 mg/kg body weight. We induced diabetes mellitus in the remaining 36 rats using an intraperitoneal injection of 1% streptozotocin (STZ) solution dissolved in citric acid-sodium citrate buffer (pH 4.5). The rats were fed 30 min after the injection. Tail blood was then drawn to measure their blood glucose levels and rats with blood glucose less than 16.7 mmol/L were excluded.

Experimental grouping

Our diabetic model was successfully established and the experimental period lasted for 20 weeks.

Twelve normal rats were included in group I-NC. Thirty-six diabetic rats were randomly divided into the II-DM group, III-DM + CW group, and IV-DM + Gli group, with 12 rats in each group. All groups were acclimatized for 1 month with normal rodent chow diet and water ad libitum prior to the start of the study. After 1 month, all groups were treated as follows for the remainder of the 20 weeks:

Group I-NC: n = 12; Normal rodent chow diet and water ad libitum

Group II-DM: n = 12; Normal rodent chow diet and water ad libitum

Group III-DM + CW: n = 12; Normal rodent chow diet and CW ad libitum

Group IV-DM + Gli: n = 11; Normal rodent chow diet and water ad libitum + daily oral Gli at dose of 0.6 mg/kg (Preetha, Girija Devi & Rajamohan, 2013).

Experimental detective index

General state

We restricted access to food and water for 8 h every 4 weeks and then weighed the rats and drew tail blood samples to measure the fasting blood glucose from each subject.

Imaging examination: Optical coherence tomography (OCT)

We examined the left eye of each rat, in each group, by OCT in the 8th, 12th, 16th, and 20th experimental weeks. The rats were weighed and then injected intraperitoneally with a mixture of ketamine (80 mg/kg) and xylazine (10 mg/kg). The left eye was dilated using tropicamide eye drops and the corneal surface was treated with sodium hyaluronate eye drops to keep the cornea moist. The entire body of the test subject was wrapped with a sterile cloth after being placed in an anesthesia-induced coma. The subject was laid on the OCT examination table so that both eyes could be examined to evaluate if the anterior and posterior diameters of the eyes were consistent with the scanning light source. The OCT examination was conducted by a second researcher and all examinations were performed by the same deputy chief physician.

We manually measured the distance from the inner limiting membrane to the retinal pigment epithelium (the thickness of the retina) using the OCT detection system. The central part of the posterior pole of the retina in rats involves the optic disc, so the total retinal thickness was measured at the disc diameter (DD) of the upper, lower, nasal, and temporal discs of the optic disc to facilitate an accurate measurement. The mean value of the four data points was taken as the total retinal thickness. Diabetes mellitus in the subjects led to cataract formation, which affected the signal strength of the OCT. The examination in all groups was terminated when half the rats in a particular group were unable to have a sufficient signal strength to perform OCT, to allow for comparison between groups.

Tissue processing/photomicrography

The rats were anesthetized as mentioned above prior to sacrifice by cervical dislocation 1–7 days after laser-treatment, at the end of the experimental period in the 20th week. The left eyeball was removed and quickly placed in an eyeball fixative solution (formaldehyde stock solution:glacial acetic acid: 95% ethanol:distilled water = 2:1:10:7) for 24 h, samples were then embedded in paraffin and cut into 4-µm-thick continuous sheets.

HE staining: Paraffin specimens were dewaxed with xylene, hydrated with gradient ethanol, long-form hematoxylin and eosin (HE) stained, and ethanol dehydrated. Neutral gum sealing was performed after the xylene was transparent. An optical microscope was used to observe the thickness of the retina and the shape and number of cells in each layer.

VEGF immunohistochemical staining: We performed a conventional dewaxing of the sample and then immersed the sample in citric acid buffer solution (pH = 6.0). We used the heat-induced antigen repair method with a microwave oven after selection. Normal goat serum was used as a blocking solution, and the rabbit anti-mouse VEGF antibody (1:100) was added at 4 °C and left overnight. After rewarming, the enzyme-labeled goat anti-mouse/rabbit IgG polymer was added dropwise and incubated at room temperature for 15 min. The specimen was washed and then incubated in the working solution for 15 min at room temperature. The sheet was counterstained, dehydrated, cleared, and sealed after DAB staining.

The expression of VEGF-positive cells in the retina of each group of rats was observed under light microscope. The positive expression of VEGF in the retina was yellow-brown in color.

Statistical analysis

The experimental data were analyzed and compared using SPSS 20.0 statistical software (IBM Corp), and the experimental data of each group were expressed by means SD. A single sample K-S test with a nonparametric test was used to verify the normality of the data. A single factor analysis of variance was used for multigroup statistical method to measure the normally distributed data. An LSD-t test was used for two-way comparison between groups. The Kruskal–Wallis H test was used for multigroup analysis for measuring the data of normal distribution, and the Mann–Whitney U test was used for intergroup analysis. The test level was set at alpha=0.05. The difference was statistically significant when the p values were <0.05.

Results

Diabetes model

Two rats were excluded because of hyperglycemia. The success rate of the established DM model was 94.44% and included rats whose blood glucose levels did not reach the standard and died within three days after STZ injection. Thirty-four rats successfully established DM. To ensure the same number of rats in each group, one rat was randomly excluded by number table. The remaining rats were divided into three groups: group II-DM, group III-DM +CW, and group IV-DM + Gli. One rat from group I-NC was excluded using the same method, and 11 rats were guaranteed in each group.

General state

Fasting blood glucose

The fasting blood glucose of rats in group I-NC was always at a normal level (4–6 mmol/L) during the experimental process and the remaining three groups had significantly higher levels. The difference was statistically significant (P < 0.05). The fasting blood glucose of group II-DM gradually increased during the 4th experimental week and maintained high levels, reaching 31.41 (±2.64 mmol/L) by the 20th experimental week.

However, the blood glucose levels of groups III-DM+CW and IV-DM+Gli gradually decreased. The blood glucose levels of groups III-DM+CW and IV-DM+Gli decreased to 24.01 ± 1.05 mmol/L and 24.25 ± 1.15 mmol/L, respectively, in the 20th experimental week. The values were far higher than group I-NC, but were lower than group II-DM, and the difference was statistically significant (P < 0.05), (Fig. 1).

Figure 1 Comparison of fasting blood sugar at different time points in rats of different groups.

Fasting blood glucose (FBG) levels of the group I NC were significantly lower than those of diabetic groups. FBG was gradually increased after the 4-week feeding experiment in group II-DM. Conversely, groups III-DM+CW and IV-DM+Gli showed a trend towards decreasing.

Body weight

There was no significant difference in the initial body weight of rats among the four groups (P > 0.05). During the experimental period, the rats in group I-NC showed a steady increase in weight when compared to the other three groups, while the weight of the rats in the remaining three groups decreased by varying degrees, showing no significant difference among the three groups (P > 0.05). In the 4th experimental week, the drinking water of group III-DM+CW was replaced with CW. Glibenclamide was administered until the 20th experiment week in group IV-DM+Gli, and there was still no significant difference in the body weight among the three groups (P > 0.05) (Fig. 2).

Figure 2 Weight comparison of rats in each group at different time points.

Group I-NC showed a steady increase in weight compared with the other three groups (P < 0.05). Conversely, groupII-DM, III-DM+CW and IV-DM+Gli showed a trend towards decreasing in weight.

OCT images of rat retinas

More than half of the rats in the II-DM group developed severe cataracts in the 16th experimental week, so the OCT signal strength was insufficient for examination. The examination was terminated in all groups. Therefore, only the results of experimental weeks 8 and 12 were obtained. The SD rat’s retina does not have a macula, but its layers have the same structure as a human’s and OCT allows for the assessment of individual layers of the retina (Fig. 3). The results in the 8th experimental week revealed that the mean of total retinal thickness of group II-DM was significantly increased, and the difference was statistically significant when compared with the other three groups (P1 = 0.001, P2 = 0.002, P3 = 0.000 <  0.05, respectively). No significant difference was observed among the other three groups (P = 0.201 >  0.05).

Figure 3 Corresponding relationship between OCT examination result and pathological section of rat retina (the range of blue arrow indicates total retinal thickness, HE *400).

(A) The OCT examination result of rat retina. (B) A pathological section of the retina of rats. OCT images and pathological findings of each layer of the retina can be seen in the two images. Blue arrows indicate the total retinal thickness. NFL: Nerve Fiber Layer; GCL: Ganglion Cell Layer; IPL: Inner Plexiform Layer; INL: Inner Nuclear Layer; OPL: Outer Plexiform layer; ONL: Outer Nuclear layer; IS/OS: Inner Segment/Outer Segment; RPE: Retinal Pigment Epithelium.

The total retinal thickness of group II-DM was the highest among the three groups over time, showing a significant difference among the other three groups (P1 = 0.002, P2 = 0.001, P3 = 0.006 < 0.05). No significant difference was observed among the other three groups at the 12th experimental week (P = 0.824 >  0.05, Table 1, Fig. 4).

Table 1 Comparison of mean total retinal thickness at different time points in rats of each group.

			Group III-	Group	
	Group I-NC	Group II-DM	DM+CW	IV-DM+Gli	
The 8th experiment week	213.63 ± 3.41	233.36 ± 12.70*	208.73 ± 9.17	209.91 ± 5.66	
The 12th experiment week	211.72 ± 4.31	235.85 ± 9.51*	209.88 ± 8.27	209.57 ± 9.27	
Notes.

* P < 0.05, Compared with Group I-NC.

** P < 0.05, Comparison with Group II-DM.

Figure 4 At the 12th experiment week of each group, OCT was used to measure the 1DD images from the superior, inferior, nasal and temporal sides of the optic nerve.

(A–D) Group I-NC, (E–H) Group II-DM, (I–L) Group III-DM+CW, (M–P) Group IV-DM+Gli.

HE staining image of rat retinas

The retinal nerve fiber layer and the outer subordinate layer remained very thin in group I-NC. The inner plexiform layer was the thickest, the inner granular layer and the outer granular layer cells were arranged neatly and connected tightly, and the structure of each layer of the retina was clear with no abnormalities.

However, in group II-DM, the inner and outer granular layers of the retina were slightly thinner, the number of cells was decreased, the arrangement was disordered, and no other obvious abnormalities were observed. The granular layer thickness of groups III-DM+CW, IV-DM+Gli and group I-NC was similar and was arranged in a disordered manner with no obvious abnormalities (Fig. 5).

Figure 5 HE staining image of retina *400 in each group of rats.

(A) Group I-NC, (B) Group II-DM, (C) Group III-DM+CW, (D) Group IV-DM+Gli.

VEGF immunohistochemical staining of retinal sections of rats

In the rat retina, VEGF protein positive expression is shown by a yellow-brown color and was mostly expressed in the cytoplasm and membrane. In group II-DM, VEGF was almost fully expressed in the ganglion cell layer and inner granular layer, and partially expressed in the outer granular layer. In group I-NC, VEGF was mainly expressed in the ganglion cell layer and the inner layer, and the color was lighter. Group III-DM + CW and group IV-DM + Gli demonstrated similar VEGF expression as in group I-NC, appearing mainly in the ganglion cell layer and inner plexiform layer (Fig. 6).

Figure 6 VEGF immunohistochemical staining of retinal sections ×600 in each group of rats.

(A) Group I-NC, (B) Group II-DM, (C) Group III-DM + CW, (D) Group IV-DM + Gli. Blue arrows indicate intraretinal granular layers, and yellow arrows indicate extraretinal.

Discussion

Our study showed that CW may have the potential to reduce blood glucose and diabetic retinal damage, but has no effect on weight change caused by DM. Coconut is a very common tropical plant. CW can be used as a potential candidate nutrient for treating diabetes and its complications, and has broad prospective applications.

In this study, insulin-dependent diabetes was successfully induced in rats by injecting STZ into the abdominal cavity to destroy the pancreatic islet beta cells. Typical diabetic changes such as elevated blood glucose and weight loss occurred during the experimental cycle. CW did not promote weight loss in rats but it partially reduced the blood glucose. Blood glucose was not reduced to normal in this study group, but it was significantly lower than that in the diabetic group. The hypoglycemic effect was similar to that of glibenclamide, which is the standard drug for DM. Our results were similar to those of Preetha, Girija & Rajamohan (2013). Previous animal experiments have found that CW can reduce blood glucose in alloxan-induced diabetic rats (Pinto et al., 2015). During pathological examination, CW was found to reduce pancreatic injury and stimulate islet beta cell regeneration in diabetic rats (Preetha, Girija Devi & Rajamohan, 2013). These results suggest that CW can effectively reduce blood glucose in diabetic rats, but its mechanism remains unknown and requires further experimental studies.

The pathophysiological mechanism of DR is very complex. In the early stages, the inflammatory damage of endothelial cells, apoptosis, and the shedding of pericytes and endothelial cells is caused by leukocyte adhesion. The reduced endothelial tight junction protein and thickened basement membrane causes damage to the inner blood–retinal barrier. This causes a disturbance in the glial inner layer of the retinal nerve and the functionality of the blood vessels, increasing the permeability of the blood vessels, and leakage of the retinal tissue (Chibber et al., 2007; Joussen et al., 2001; Nag, Kapadia & Stewart, 2011; Tang & Kern, 2011; van Hecke et al., 2005). OCT may detect the changes in the retinal thickness caused by retinal tissue leakage at an early stage and assist in providing quantitative information. HE staining on the retinal section of rats revealed the extent of damage to the layers of the retina at the cellular level. In this experiment, the increased thickness of the retina was observed in the 8th and 12th experimental week in rats with diabetes, suggesting that retinal tissue leakage occurred in the early stages.

We were unable to obtain data for the next few weeks of the experiment. The retinal thickness in groups II-DM + CW and IV-DM + Gli was the same as that in normal rats. HE staining of the retinal sections revealed that in group II-DM, visible pathological changes were noticed in the retina at the 20th week, such as thinner inner and outer granular layers reduced the number of cells and disordered arrangement.

However, these changes were not observed in the retinal layers of group II-DM+CW. These results indicated that CW may reduce cellular damage in various layers of the retina and reduce leakage of retinal tissues.

VEGF is currently the most important and directly stimulating angiogenic factor. It has strong vascular permeability, can significantly increase microvascular permeability, and destroy BRB (Praidou et al., 2010). Various angiogenesis-related factors promote the proliferation and permeability of microvascular endothelial cells by directly or indirectly inducing the expression of VEGF and its receptors (Rakhila et al., 2016). VEGF is closely related to retinal tissue leakage.

VEGF immunohistochemical observation was performed on the retina of rats. In the 20th experimental week, VEGF was almost fully expressed in the ganglion cell layer, inner plexiform layer, and granular layer in the II-DM group, and was partially expressed in the outer granular layer. The retinal VEGF was mainly expressed in the ganglion cell layer and the inner layer in group III-DM + CW with a lighter color; the appearance was similar to that found in group I-NC. We should seek to find how CW can reduce the retinal VEGF expression level, affecting retinal vascular permeability and reducing retinal tissue leakage.

Diabetes may cause hyperglycemic episodes which in turn impacts five key biochemical pathways: the polyol pathway activation; the production of advanced glycation end products (AGEs); protein kinase C (PKC) activation; hexosamine pathway activation; and poly (ADP-ribose) polymerase upregulation. Several DR mechanisms have been shown to be involved in the overexpression of VEGF. Activated protein kinase C (PKC) can promote an increase in VEGF expression levels (Amadio et al., 2012). The upregulation of the expression of VEGF is affected by the formation of advanced glycation end-products (AGEs) and the activation of the receptor for advanced glycation end-products (Kandarakis et al., 2014; Yan, Ramasamy & Schmidt, 2010). The five key biochemical pathways lead to oxidative stresses, resulting in mitochondrial dysfunction, deregulation of proinflammatory mediators, and hypoxia. These effects cause the apoptosis of vascular and neuronal cells and the upregulation of VEGF expression. The generation of ROS and oxidative stress further exacerbates metabolic dysfunction, leading to elevated ROS production in a self-perpetuating positive feedback mechanism (Wang et al., 2017).

This reveals the important role of antioxidants in DR. CW is a natural beverage obtained directly from the cavity of the coconut fruit. It has a very high nutritional value and acts as a natural antioxidant. Clinical trials revealed that CW may increase antioxidant enzymes and reduce lipid peroxidation (Santosa, 2015). Many oxygen free radical scavengers, such as ascorbic acid, cysteine, phenolic compounds, and L-arginine are present in CW and protect organs via their antioxidant capabilities (Amadio et al., 2012a; Rakhila et al., 2016). CW may act to remove ROS and reduce OS, reduce the activation of PKC and formation of AGEs, decrease the expression of VEGF, and decrease the leakage of retinal tissue through its antioxidant properties. However, the specific mechanisms of action require further exploration.

Conclusions

CW has the potential to reduce blood glucose and diabetic retinal damage but it has no effect on weight change caused by DM. We speculated that this may be due to its antioxidant properties. Further studies are needed to investigate the role of coconut water supplementation in DM and which antioxidants in coconut water may be used to develop candidate drugs.

Supplemental Information

Supplemental Information 1 Supporting information for experimental manipulation

Click here for additional data file.

Supplemental Information 2 Raw numbers of fasting blood sugarbody weight and oct

Click here for additional data file.

The authors appreciate the Central Laboratory and Pathology, Department of Haikou Hospital affiliated with Xiangya Medical College of Central South University for providing professional laboratory, testing equipment and professional guidance.

Additional Information and Declarations

Competing Interests

Author Contributions

Animal Ethics

Data Availability

The authors declare there are no competing interests.

Yanan Dai, Li Peng and Xiaohua Zhang conceived and designed the experiments, performed the experiments, authored or reviewed drafts of the paper, and approved the final draft.

Qingjing Wu performed the experiments, authored or reviewed drafts of the paper, and approved the final draft.

Jie Yao, Qiu Xing and Yunyan Zheng analyzed the data, authored or reviewed drafts of the paper, and approved the final draft.

Xiaobo Huang analyzed the data, prepared figures and/or tables, authored or reviewed drafts of the paper, and approved the final draft.

Shaomei Chen performed the experiments, analyzed the data, prepared figures and/or tables, authored or reviewed drafts of the paper, and approved the final draft.

Qing Xie conceived and designed the experiments, authored or reviewed drafts of the paper, and approved the final draft.

The following information was supplied relating to ethical approvals (i.e., approving body and any reference numbers):

Feeding, operation and treatment of all experimental animals met the requirements of the biomedical ethics committee of Haikou People’s Hospital (2017-087).

The following information was supplied regarding data availability:

Raw data are available as a Supplemental File.

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
