# Peer review of "Effects of coconut water on blood sugar and retina of rats with diabetes"

_PeerJ, doi:10.7717/peerj.10667_

## Round 0.1 · original submission · Major Revisions

There are two different titles given in the manuscript. I prefer the title on the first page.

The title on Line 1 appears is an error. Coconut water may have a hypoglycemic effect, but the experimental rats were hyPERglycemic, and remained hyPERglycemic despite coconut water. (Figure 1)

An English language editor should review the entire article. If the authors have a word processing program such as Microsoft Word, spell check should be performed prior to the next submission.

There are numerous grammatical errors e.g. Line 34 had versus has?

Line 39, Line 62, Line 97, Line 111, Line 123 (restricted rather than forbidden), Line 124, Line 133, Line 147 (wrong tense)

Line 99 pH versus Line 101 PH Line 277 preetha versus Preetha.

Abstract Background: Was any other “general condition” factor other than weight analyzed

Line 23: Use the long form of VEGF on first mention

Line 45 type 1 or type 2 diabetes mellitus

Line 50: Summarize the current theories of pathogenesis in 2-3 sentences.

AntiVEGF and corticosteroid injections/implants are extensively given for diabetic retinopathy

Line 83 Were green coconuts chosen, as they have more water?

Explain that glibenclamide is a hypoglycemic that stimulates pancreatic beta cells. In

Provide the normal rat blood sugar level.

Line 155 Use long form hematoxylin and eosin on first usage of HE

Line 178 Correct punctuation of both statistical tests

Line 184 Were not 36 rats assigned to the experimental group? 34/36 =94.4%

Line 221 Unlike humans rats have no fovea and in human diabetic studies one would concentrate on the macula rather than the peripapillary area.

Was there any missing data? If so, how was it handled?

Figure 3: I am more familiar with the layers of the human retina. Are there any extra descriptors to “outer membrane”. Do rats have an outer plexiform layer?

On the second last page, underneath Table 1, the same sentence is repeated twice.

Reviewer 1 ·

Basic reporting

The article would benefit from an English language editor who should review the entire article to improve flow, grammar and ambiguous language.

Line 39- Unsure of how non-communicable disease relates to this article. Simply put Diabetes is one of the most significant public health problems worldwide.

Line 40-Authors could substitute epidemiological data from global data of incidence of DR and PDR as rat model not specific to DM in Chinese patients

Line 44- DR is one of the microvascular complications of DM and the major ocular complication.

Line 47-The authors may want to note DR is the most common cause of vision loss in the working population

Line 50-The statement that the pathogenesis of DR is unclear is misleading. The authors should expand on the role of hyperglycaemia, pericyte death and inflammatory cytokines such as VEGF while noting limitations. Additionally there are specific limitations of animal models which may further put this article into context. (Kern T, S, Antonetti D, A, Smith L, E, H: Pathophysiology of Diabetic Retinopathy: Contribution and Limitations of Laboratory Research. Ophthalmic Res 2019;62:196-202. doi: 10.1159/000500026)

Line 51-The authors should note the role of anti-VEGF therapy (Protocol S) as well as, steroids, PRP and focal laser. The role of surgery is for the complications of proliferative diabetic retinopathy such as vitreous hemorrhage and tractional retinal detachment. The authors may want to expand on glycemic control and blood pressure control in prevention.

Line 52 “Early effective drug treatment is still in the exploratory stage, and so it is very important to find an effective drug treatment to prevent in the early stage of onset”…This statement is misleading.

Lines 55-63…The authors should simplify this part of introduction to state CW is abundant….some promise in regulating blood sugar, carbohydrates and reducing microvascular complications of neprhopathy…The authors wondered if there would be a role in reducing DR, another microvascular complication.

Line 64-The whole body condition is ambiguous…the authors should study what was measured and why.

Line 69- Add units to rat weight in manuscript and excel data file. Change refractive stroma to corneal stroma

Line 117- Could rephrase to “Based on an adult body weight of 60 kg, a typical initial dosage of glibenclamide would be 5 mg per day”

Line 138- The total retinal thickness was defined as the the distance between the ILM and RPE. These are relatively hyper reflective…but this statement is unnecessarily complicated.

Line 144- Could be rephrased for clarity “Diabetes mellitus in rats led to the formation of cataract which affects the signal strength of the OCT. When half the rats in a particular group were unable to have a sufficient signal strength to perform OCT, examinations in all groups were terminated to allow for comparison between groups.”

Line 193-195- The wording of these lines are clumsy. “the fasting blood sugar of group I-NC was always at normal  level (4-6 mmol/L), while the other three groups had significantly higher levels than group I (P>0.05)”

Line 206-210- These sentences could be misleading, please indicate that although there was no significant difference there was a trend for bodyweight.

Line 216-19-These sentances are confusing “Only results at week 8 and 12 were obtained as at week 16 over half of group II-DM rats had a lens opacity precluded adequate signal strength of the OCT”

Lines 221-223. Language is confusing, could be rephrased as “OCT allows for the assessment of individual layers of the retina”

Line 283- The pathophysiology of DR is complex but I would avoid describing it as very complicated.

Experimental design

Line 69- make the age of the rats less ambiguous “about 10 weeks”
The temperature range is not ideal many labs would store at 22+/- 2 Celsius. Hypo and hyperthermia may affect the glycemic control and caloric requirements of the rats.


Line 83- Using local coconuts may be convenient but may make it challenging to reproduce the results as the concentrations of sugar, vitamins, minerals, amino acids and other factors may differ between crops. An analysis of the nutrition of water fed to rats may have been helpful.

Although OCT was unable to be performed due to cataract formation beyond week 16, it would be interesting to determine the effect of coconut water in diet on the formation of visual significant cataract.

Line 167- Why did the authors choose 4 fields of view for this analysis? Additionally how were the fields chosen? Were they chosen at random and averaged? Although there is no standard others have sampled 10 areas at random and averaged. More sampled areas would likely give more reproducible results.
(Jun PENG, Kun PAN, Zheng-Rong LIU, Yu-Hui QIN, Qing-Hua PENG,
Effects of Shuang Dan Ming Mu Capsule on Expression of VEGF-a, VEGF-b, VEGF-c and the VEGF receptor, Flk-1, in Diabetic Retinopathy Rats,
Digital Chinese Medicine, Volume 1, Issue 3, 2018,Pages 228-238,)

Validity of the findings

Line 267-Retinal thickness is not an accurate surrogate for diabetic retinopathy. It is an early change in DR, however you may expect to see early microvascular changes in DR. It would have been useful if the authors by examining the microvascular changes in the contralateral eyes looking for pericyte density and considered a trypsin digest. Their rats were sacrificed at 20 weeks and likely would have demonstrated some microvascular changes. This would have strengthened their findings and allowed for a comment on DR. The authors may be limited as the experiments have already been completed and should discuss limitations related to their limited OCT, histologic and immunohistologic analysis.

Line 269-Coconut water is a natural product, unless the authors plan to assess specific molecules within coconut water they should avoid discussing coconut water as a candidate drug. Instead they should discuss its role as a nutritional supplement.

The authors assessed VEGF expression in the rat retina, however assessment of specific isoforms as well as other cytokines known to be associated with DR would have added depth to their experiments.

The conclusions are misleading and should focus on what this study demonstrated….fasting blood sugar, retinal thickness as measured by OCT and retinal VEGF expression at 20 weeks weeks were all lower in diabetic rats who consumed coconut water. We speculated this may be due to its antioxidant properties. Further studies are needed to investigate the role of coconut water supplementation in DM and which antioxidants in coconut water may be developed into candidate drugs.

Additional comments

My understanding is that the diabetic model was induced at 10 weeks age and then rats were sacrificed at week 20 of the experiment. It may be useful in the figures and text to clarify week of experiment vs week of age.

Figure 3- Although Inner and outer nuclear layers can also be referred to as granular layers, it may be best to refer as inner and outer nuclear layers as these are the preferred terminology for OCT interpretation. The resolution of the OCT scan is not ideal in this image, if possible use an image with a higher DPI and more clarity to allow for definition of the retinal layers. Additionally the rat retina should demonstrate NFL, GCL, IPL, INL, OPL, ONL, ELM, IS/OS, RPE and choroid. It may be better to use this terminology or at least label the OCT and pathology slide separately.

The authors discussion of DR and VEGF is oversimplified and they should briefly discuss the clinical pathophysiology (hyperglycaemia and pericyte death) before discussing the molecular pathophysiology where they discuss PKC, AGEs and ROS. However they do not mention the hexosamine pathway, polyol pathway, poly (ADP-ribose) polymerase activation, renin-angiotensin aldosterone system and inflammation. For a review they authors my look at (Michael Whitehead, Sanjeewa Wickremasinghe, Andrew Osborne, Peter Van Wijngaarden & Keith R. Martin (2018) Diabetic retinopathy: a complex pathophysiology requiring novel therapeutic strategies, Expert Opinion on Biological Therapy, 18:12, 1257-1270, DOI: 10.1080/14712598.2018.1545836)

·

Basic reporting

The English needs major revision throughout the paper. Multiple grammar, syntax, and punctuation errors.
The introduction is sufficient, however, some references are missing
Conforms to PeerJ article structure; however, it requires significant revision with regards to the language.
Only blood glucose and weight data are shared - OCT and histology data are not made available

Introduction:
1- Line 46: Reference needed for the incidence of DR in Chinese DM patients.
2- Line 50: Authors claim that DR pathophysiology is still unclear; The authors should state that it has been previously studied and is multifactorial (possibly include a short summary with relevant references)
3- Line 50: Clarification needed for the treatment options discussed. What types of laser/surgery, accompanied by relevant references.
4- Line 51: Reference needed for the claimed “poor prognosis”.

Figures 1-2: Statistical significance should be shown by a notation. It should be clarified what error bars represent (SD, SE, CI,..). The abbreviations in figure legends should be explained in the caption.
Figures 4-5: Areas of interest should be highlighted with an arrow (similar to Fig 6)
Figure 7 is likely not necessary, as this is not a review paper.

Table 1 - The abbreviations in figure legends should be explained in the caption; values at baseline and week 4 should be included and statistically compared (to be included in the results section as well)

Experimental design

The article fits the scope of the journal.

Methods:

- Sample size analysis to be included in the methods section.
- Although a sample size of 48 is powered (at alpha 0.05, B 0.20, small effect size 0.5) for the majority of the analyses performed by the authors, their OCT analysis is underpowered as obtaining quality OCT images was not possible in more than half of group II at 12W. The potential loss of sample through the experiment should have been accounted for in the power analysis and authors should have started with a large sample. At this point, this limitation should be mentioned in the discussion.
- The authors decided to stop OCT measurements once images were not obtainable in half of the rats in one group. This occurred at 12W in group II. However, authors could have continued obtaining OCT images in groups I, III, IIII at subsequent time points.

Line 69: The sentence should be rephrased.
Line 69: Change to Sprague-Dawley (SD)
Line 69: the units for the weight is missing
Line 103: Rats with glucose <16.7 were excluded (is it made clear in results how many? )
Line 113: Authors describe Group III as “normal rats” which contradicts their previous description of group III (diabetic rats). Such inconsistencies appear throughout the manuscript.
Line 117: “the adult body weight was 60 kg”. Unclear what authors are referring to.
Line 119: Conversion formula should be included and referenced.
Line 151: Laser treatment? – to be explained
Line 169: “Four specimens were taken from each specimen”. To be revised

Validity of the findings

Described statistical methods are sound, however, the reported results are mainly descriptive and p-values are missing at many places. More details below:

Results:
- All means and standard deviations should be included in the text.
- Precise p-values should be disclosed.
- It is unclear how many mice underwent OCT at weeks 8 and 12, at each group. Authors state that more than half of group II could not undergo OCT at Week 12; however, it should be further clarified specific to each group and each time point
- The authors should state the baseline total retinal thickness and compare across groups.
- In addition to increased retinal thickness, authors should state if any developed cystoid edema.
- Authors should report whether there were any significant differences in the fasting blood sugar level of each group prior to the commencement of the experiment/induction of DM
- Also, what is shown as week 0 in Fig 1 and 2, can be misinterpreted as pre-DM induction. I suggest that authors add the pre-DM induction values for all variables of interest to the results (and compare across the groups) and add them to the figures (as baseline). Subsequently, rename what is currently labeled as “0w” to 72h post DM induction (if that is truly what 0w is indicating).

Line 184: If 34 rates were included, success shall be 34/36= 94.4%.
Line 194: authors simply state that the fasting blood sugar was significantly higher in DM groups compared to the control. However, they state the p-value as P>0.05. Additionally, the average values shall be stated.
Line 196: Authors state that from week 4, the blood sugar increased in group II, however, in the absence of baseline values, it is difficult to apprehend the degree of increase. Further, no statistics are presented here. The same comment applies to lines 219 to 222.
Line 202: same comment as above; “P>0.05”.
Line 208: The authors report increases or decreases in weight in each group. It should be clarified to what extent these changes were, and whether they were significant.
Line 211: “glibenclamide was initiated until week 20”. It seems like week 4 is left out of the sentence. To be revised.
Line 226: Total retinal thickness values should be included in the text
Line 239-242: Authors should quantify the thickness rather than simply resorting to subjective descriptions.
Line 244: “slightly thinner”, “number of cells was decreased”, the author should clarify the comparison group and verify the statistical significance (with p-values)
Line 261-263: “p=0.00” does not make sense. Should revise (?p<0.001)

Discussion:
Line 281: The authors claim that the coconut water can effectively reduce blood sugar in diabetic rats and state that the mechanism is unknown. They should include a few theories/hypotheses about the possible mechanisms.
Line 328-336: Authors attempt to address possible pathways through which coconut water can decrease VEGF expression. It is also possible that lower blood sugar levels limited VEGF over-expression, to begin with.

---

## Round 0.2 · Minor Revisions

The article still requires review by an English language editor.

Reviewer 1 ·

Basic reporting

I recommend again that an English language editor review the entire article.  The authors track changes document identify many errors which can rapidly be revised. There are numerous new errors in spaces, tense and flow after incorporating the revisions.

Experimental design

I was very pleased with the rebuttal by the authors and feel their methods are justified.

Validity of the findings

No comment

Additional comments

I am pleased with the authors rebuttal and incorporation of revisions to the text. I think the experimental design and validity of the findings are appropriate for publication in the journal but not in its current form as the authors have many errors in punctuation, grammar, syntax, and flow that could rapidly be improved by an English language editor revising the document. This was recommended to them in the last review. I recommend they do this.

---

## Round 0.3 · accepted · Accept

Line 153. Correct the spelling to tropicamide